# Pesticide Exposure in the Cultivation of *Carica papaya* L. and *Capsicum annuum* L. in Rural Areas of Oaxaca, Mexico

**DOI:** 10.3390/ijerph21081061

**Published:** 2024-08-13

**Authors:** Héctor Ulises Bernardino-Hernández, Yael Gallardo-García, Gerardo Vargas-Valencia, Arturo Zapién-Martínez, Gabriel Sánchez-Cruz, Leobardo Reyes-Velasco, José Ángel Cueva-Villanueva, Ericel Hernández-García, Jaime Vargas-Arzola, Honorio Torres-Aguilar

**Affiliations:** Chemical Sciences Faculty, Benito Juárez Autonomous University of Oaxaca (UABJO), Av. Universidad S/N. Cinco Señores, Oaxaca de Juárez, Oaxaca C.P. 68120, Mexico; yael.07@hotmail.com (Y.G.-G.); geravar28@outlook.com (G.V.-V.); zaarma@yahoo.com (A.Z.-M.); gsanchez@uabjo.mx (G.S.-C.); lreyes.cat@uabjo.mx (L.R.-V.); angelus_pub@hotmail.com (J.Á.C.-V.); ericledimp@gmail.com (E.H.-G.); vajcquabjo@hotmail.com (J.V.-A.); qbhonorio@hotmail.com (H.T.-A.)

**Keywords:** agricultural production system, papaya, chili, pesticide exposure, acute poisoning symptoms, health risk

## Abstract

This study focuses on describing the diversity of pesticides, the knowledge and behaviors of their use, and the acute poisoning symptoms (APS) derived from their exposure from two agricultural production systems (papaya—*Carica papaya* L.- and chili—*Capsicum annuum* L.-) in Oaxaca, Mexico. Through surveys, sociodemographic information, characteristics of the production system, knowledge and behaviors in the handling of pesticides, and APS perceived by users were captured. Papaya producers are younger, have fewer years of activity, and have larger agricultural areas than chili producers. Insect attacks and diseases are an essential factor for the application of pesticides. Thirty-one active ingredients (Ais) were identified in papaya and thirty-seven in chili, predominantly insecticides and fungicides of toxicological category IV. Approximately 50% of users apply mixtures of different Ais, have little knowledge and inappropriate behavior in their handling, and report up to five acute pesticide poisoning symptoms, mainly burning and irritation of the skin, burning eyes, itchy skin, runny nose, headache, and watery eyes. The production of papaya and chili are relevant activities for generating economic income, but they risk the producer’s and their family’s health. Both systems are a potential scenario for the manifestation of diseases due to exposure to pesticides in the medium and long term.

## 1. Introduction

Agricultural activities worldwide have been fundamental to many countries’ economic strength and growth, particularly in developing countries [1]. In the case of Mexico, this sector has suffered a decrease in its contribution to the national GDP, from 4.4% in 1995 to 3.4% in 2017 [2]. Despite this decrease, these activities continue to be a sector that generates employment and income in rural and urban areas of Mexican territory. The northwestern states are characterized by presenting the most significant technological and industrial development, as well as large areas of agricultural cultivation for commercial and export purposes, highlighting Sinaloa, Jalisco, Tamaulipas, Guanajuato, Michoacan, and Sonora in the production of grains (rice, barley, beans, corn, sorghum, and wheat), while Veracruz, Michoacán, Tabasco, Tamaulipas, Jalisco, and Colima stand out for the production of fruits (avocado, cocoa, coconut, strawberry, guava, lemon, mango, apple, orange, walnut, papaya, pineapple, and banana) [3]. For the rest of the country, agricultural products are used to a lesser extent for commercial purposes and to a greater extent for subsistence.

During the last decades, the performance and productivity of agricultural activities in Mexico have been maintained and even increased through conventional management, including the intensification of natural resources (soil and water), machinery, and the use of chemical inputs, highlighting fertilizers and pesticides. Regarding the latter, in 2021, Mexico ranked twentieth in the agricultural use of pesticides, mainly applying fungicides, followed by herbicides and insecticides. The countries that occupied the first places were Brazil, the United States, Indonesia, China, and Argentina [4]. Between the period 2000 and 2020, the total consumption of pesticides in Mexico increased between 57 and 65% [5]. The National Institute of Ecology and Climate Change of Mexico points out that the most widely used chemical groups reported in Mexican territory are organophosphates and carbamates [6].

The damage to human health caused by exposure to pesticides is due to the chemical group, half-life, and toxicity to which the various active ingredients that make them up belong, which generates effects immediately (acute poisoning) or in the medium and long term (chronic poisoning) [7]. Among the main acute poisoning symptoms are nausea, abdominal pain, vomiting, and diarrhea in the gastrointestinal system; chest pain, cough, and shortness of breath in the respiratory system; dizziness, headache, confusion, profuse sweating, and ataxia in the nervous system; bradycardia, tachycardia, hypertension, and hypotension in the cardiovascular system; muscle weakness, pain, and stiffness in the muscular system; swelling, itching and irritation of the skin, and tearing, pain, irritation and itching of the eyes, among others [8]. Regarding the long-term effects derived from chronic exposure to pesticides, their carcinogenic and neurotoxic power has been associated with different alterations of the renal, reproductive, immunological, endocrine, hepatic, and neurological systems, among others [9,10,11,12,13].

These inputs have permeated rural areas among small-scale and subsistence producers in the Mexican southeast. Such is the case of the State of Oaxaca, where the main self-consumption crops are corn and beans. In contrast, the most prominent commercial crops are coffee, sugar cane, lemon, mango, sesame, orange, banana, agave, papaya, avocado, and chili [14]. Mexico is one of the leading producers of papaya internationally, and Oaxaca is one of the leading states in national production, mainly in the Costa region, due to its geographical and climatological conditions [15]. Chili is one of the most cultivated vegetables worldwide, and Mexico is one of the leading producers [16]. Although Oaxaca is not a significant state leader in producing this vegetable, its cultivation is relevant since it is part of the state’s economic activities. In particular, various farmers from the town of José María Morelos in the municipality of Santa María Huazolotitlán, in the Costa region, are dedicated to growing papaya, while other producers in the community of Arroyo Limón, municipality of San Juan Bautista Tuxtepec, in the Papaloapam region, are dedicated to the cultivation of chili. Both production systems have similar climatic conditions. The workforce is family-based and is characterized by conventional agricultural management through applying chemical pesticides to prevent and control pests and diseases in their crops. This has generated a scenario where producers expose their health and that of their families at the expense of agricultural profitability and the generation of economic income, which is the basis of family sustenance. According to SEMARNAT [17], in Mexico from 2012 to 2017, the states with the highest incidence of new cases of pesticide poisoning were Jalisco, Guerrero, Michoacan, Chiapas, and the State of Mexico. Oaxaca is located in the eleventh position. The states of southeastern Mexico, including Oaxaca, are those with the highest mortality rates associated with acute pesticide poisoning, with self-inflicted cases predominating, followed by accidental poisoning [18]. In the case of the rural population dedicated to agricultural activities in Guerrero, 8% to 61% of acute pesticide poisonings have been reported [19], while in rural communities in Tabasco, the rate has been 37% [20]. By the above, it is necessary to document the diversity of pesticides and their mixtures that are used in the agricultural field, as well as the knowledge and behaviors in the use and management of said substances and the probable damage to health in the population, which is dedicated to agricultural activities. This research aimed to describe pesticide exposure and harm to health through the acute pesticide poisoning symptoms in papaya and chili producers from two rural communities in Oaxaca, Mexico.

## 2. Materials and Methods

### 2.1. Study Area

This study was carried out in rural communities of José María Morelos (JMM), belonging to the municipality of Santa María Huazolotitlán (Latitude: 16°21′15″ N and Longitude: 97°55′47″ W, 30 m above sea level, climate warm subhumid, in the Costa region) and Arroyo Limón (AL), located in the municipality of San Juan Bautista, Tuxtepec (Latitude: 17°54′01” N and Longitude: 95°57′40” W, 80 m above sea level, warm humid climate, in the Papaloapan region), in the State of Oaxaca, Mexico (Figure 1). In José María Morelos, the population is considered black Afro-Mexican [21]. It is characterized by having a papaya production system (PPS), while in Arroyo Limón, the population is of Chinanteco origin [22] and has a chili production system (CPS).

### 2.2. Sampling Procedures

The study universe was the adult male population dedicated to agriculture in both locations (JMM: total population: 2331, adult population: approx. 850; AL: total population: 1298, adult population: approx. 450). From June to December 2019, through field trips and prior authorization from local authorities, papaya and chili producers were identified and invited to participate in the study voluntarily. Individuals of legal age and those who used pesticides in agricultural activities were included. The sample included 55 producers (30 from the PPS and 25 from the CPS). The unit of analysis was each individual (producer), to whom a survey was applied to document age and schooling, characteristics of the production system (calendar of agricultural activities, pests, and diseases), use and management of pesticides (trade names, frequency of application, mixtures used, and hours of fumigation), knowledge and behavior regarding the use of pesticides and acute pesticide poisoning (APP) symptoms perceived during or after fumigation (headache, burning and watering in the eyes, irritation of the skin, nausea, dizziness, vomiting, salivation, and cramps in feet and hands, among others) [23,24]. The survey was previously piloted and validated by those responsible for the study.

### 2.3. Statistical Data Processing

Based on the trade names provided by users and observation of the label of pesticide containers in the field, these products were characterized by reviewing the technical sheets (safety sheet) to determine the chemical classification, active ingredient (AI), toxicological category (TC), and type of pesticide by its action (herbicide, insecticide, or fungicide). The toxicological category was considered based on the proposal of the Federal Commission for the Protection against Sanitary Risks (COFEPRIS), based on the LD50 expressed in mg/kg recommended by the WHO, following the Official Mexican Standard NOM-232-SSA1-20097 [25,26]. To establish adequate knowledge and behavior, answers consistent with correct knowledge and behavior were scored with one point, and those found otherwise were scored with zero points. With the estimates obtained globally, cut-off points were established based on the calculation of quartiles for each set of questions for each variable: six questions for knowledge and thirteen for behaviors. Sufficient and appropriate knowledge and behaviors were considered in quartile IV (≥three and ≥seven positive responses for knowledge and behaviors, respectively).

The information was analyzed through a frequency analysis for the qualitative variables and measures of central tendency and dispersion for the quantitative variables. Student’s *t*-test for independent samples or the Mann–Whitney U test was used to compare the production systems (PPS versus CPS) concerning the quantitative variables (age, seniority in agricultural work, sown agricultural area, total of pests and diseases, total of AI, fumigation time, scores on knowledge and appropriate behaviors in pesticide use, and number of APP symptoms), according to normality (Shapiro test) and homoscedasticity of variances (Levene test) of the data. The X^2^ test was used for the qualitative variables (producer sex, schooling, and used pesticide mixtures). Pearson’s R or Spearman’s Rho was used to establish possible associations between quantitative variables (number of APP symptoms with the number of active ingredients, time to fumigate a hectare, and planted area) depending on the data’s normality. Statistical analysis was performed using SPSS version 15.0 for Windows (IBM, Armonk, New York, USA).

## 3. Results

### 3.1. Sociodemographic Characteristics

The study’s population was predominantly male. No significant differences were observed between the production systems regarding education, and the majority know how to read and write. Significant differences were observed regarding age, seniority in agricultural work, and planted area. The PPS producers are younger, have fewer years in papaya production, and have larger agrarian areas than the CPS producers, who are older, have more time in chili production, and have small agricultural areas (Table 1).

### 3.2. Characteristics of Production Systems

In the CPS, the cultivated variety is known in the region as chili Soledad (*Capsicum annuum* L.). The agricultural cycle lasts approximately ten months and consists of ten stages. It begins in March with the preparation of the land where the plant will be grown and ends in February of the following year with the burning of post-harvest waste. The activities are carried out mainly with manual tools such as the hoe, barrow, and machete. Regarding the PPS, the main variety is known as Maradol papaya (*Carica papaya* L.). The cycle lasts approximately 12 months and consists of 13 stages. It begins in September with the selection of the land next to be planted and ends in October of the following year. Agricultural activities involve hand tools, heavy machinery, and accessories such as tractors, fallows, plows, and edgers (Table 2 and Table 3). The CPS requires a lower economic investment and labor force compared to the PPS, given that production costs are mainly related to the acquisition of fertilizers and pesticides. This is in contrast to the PPS, where the inputs and equipment used are more significant and include propagation trays, rooting agents, fertilizers, pesticides, dispersing agents, PVC pipe and irrigation tapes, pumping systems, and agricultural machinery, which increases production costs.

### 3.3. Pests and Diseases in Crops

Producers of both production systems agree that pests and diseases cause the most significant damage to their crops. In particular, the lack of money and weeds are considered dangerous to the CPS’s productivity. At the same time, in the PPS, low soil fertility and poor quality water for irrigation influence the system’s productivity (Figure 2).

The white mite and the white fly are the main insects that cause the most significant damage in both production systems. In particular, in the PPS, the black mite, red mite, and red spider stand out as highly harmful pest insects, while in the CPS, crickets, cutworms, and nematodes stand out. Regarding the diseases related to the most significant damage in both systems, there is curling of the leaves, roots, and stem rot. Specifically, mosaic and plant rot stand out in PPS, while in CPS, defoliation, ring spots, and plant rot in the nursery stand out (Figure 3). The total average of pests and diseases was statistically higher in the PPS (11.5 ± 2.6, amplitude = 7 to 16) compared to the CPS (9.6 ± 2.2, amplitude = 4 to 14) (t = 2.984, df = 53, *p* = 0.004). However, when comparing the total number of insects, no significant differences were observed between the systems (PPS = 6.2 ± 1.8, amplitude = 3 to 9; CPS = 5.7 ± 1.0, amplitude = 4 to 8; Mann–Whitney U = 279.5, *p* = 0.099), but the CPS presented a significantly higher average of diseases compared to the PPS (CPS = 9.6 ± 2.2, amplitude = 4 to 14; PPS = 5.3 ± 1.7, amplitude = 2 to 8; Mann–Whitney U = 707.0, *p* = 0.000).

### 3.4. Use of Pesticides on Crops

A total of 49 IAs were identified from 77 commercial products (Table 4). In the CPS, a greater diversity of AI and commercial products is used compared to the PPS, although no significant differences were observed in the average of AI used between the systems (PPS: 7.8 ± 2.7, amplitude = 3–13; CPS: 7.9 ± 2.6, amplitude = 4–16; Mann–Whitney U = 368.5, *p* = 0.912). In both systems, the predominance of insecticides and fungicides over herbicides and bactericides is observed. In particular, the use of mainly TC IV insecticides and fungicides stands out. In general, no significant correlations were identified between the number of AIs used and the number of pests and diseases (Spearman’s Rho = 0.144, *p* = 0.296), as well as with the planted area (Spearman’s Rho = −0.074, *p* = 0.589).

The most used insecticides in the PPS include methomyl and carbofuran (TC II), followed by malathion (TC IV). In CPS, imidacloprid (TC IV) stands out, followed by the combination of lambda-cyhalothrin + thiamethoxam (TC III/IV), the combination of imidacloprid with beta-cyfluthrin (TC IV) and abamectin (TC IV). Regarding the use of fungicides in the PPS, mancozeb, prochloraz, metalaxyl, captan, and azoxystrobin stand out (all from TC IV), while in the CPS, only mancozeb and captan (TC IV) stand out. Regarding herbicides, in both systems, the use of paraquat (TC III) and glyphosate (TC IV) predominates through products with different commercial names; in CPS, ammonium glufosinate (TC IV) is additionally frequently used. Although few producers use bactericides, in both systems, the presence of products with streptomycin and oxytetracycline was observed (Table 5).

### 3.5. Use of Pesticide Mixtures

No significant differences were observed between the systems regarding the proportion of producers who agreed to make pesticide mixtures (X^2^ = 0.973, RV = 0.976, *p* = 0.324, 60% in the PPS, and 46.7% in the CPS). Twenty different mixtures were identified in the PPS, and thirteen in the CPS. It should be noted that several producers use different mixtures. In the PPS, the mix of herbicides (mainly paraquat and glyphosate) with insecticides (mainly malathion, lambda-cyhalothrin, and carbofuran) stands out, followed by the mixture of fungicides (mainly prochloraz) with insecticides (mainly methomyl) and the mix of two different fungicides (mainly mancozeb with captan or prochloraz). In CPS, the mix of two different herbicides (paraquat with glufosinate ammonium or glyphosate) stands out, followed by the blend of mancozeb (fungicide) with other fungicides or insecticides (Table 6). No significant differences were identified between the systems regarding the time to spray one hectare (PPS: 4.5 ± 1.9 h, amplitude = 2–8; CPS: 4.5 ± 1.3 h, amplitude = 2–8; Mann–Whitney U = 406.5, *p* = 0.587). In the PPS, users allocate approximately 22 h a week to fumigate, from the transplant stage to flowering and harvest, where the hours and application of pesticides are intensified. The exposure period is approximately seven months during the agricultural cycle. In the case of CPS, producers allocate approximately 30 h a week from transplanting to the flowering stage and the last cut of chili, where the application of pesticides is intensified. The exposure period is approximately nine months during the crop cycle.

### 3.6. Knowledge and Behavior in the Use and Management of Pesticides

No significant differences were identified between the systems regarding the scores on appropriate knowledge in the use of pesticides (PPS = 1.8 ± 1.3 and CPS = 2.3 ± 1.1, respectively; Mann–Whitney U = 478.0, p = 0.071), but it was identified that the PPS producers presented the highest appropriate behavior scores (PPS = 6.7 ± 1.0 and CPS = 5.4 ± 1.1; Mann–Whitney U = 143.5, p = 0.001). In both systems, it was observed that most of the producers presented insufficient knowledge (PPS: 72.0% and CPS: 63.3%; users who presented inappropriate behaviors in the handling of pesticides (CPS: 83.3% and PPS: 44.0%; X^2^ = 9.330, RV = 9.574, p = 0.002)). Insufficient knowledge is related to the lack of professional training in the use of pesticides. Learning is based on advice provided by family and friends, and they do not read or follow the instructions or know the meaning of the color codes on the labels. Inappropriate behaviors are related to the poor protection of hands, body, and face when handling pesticides and their application, insufficient maintenance of fumigation equipment, inadequate storage of pesticides in their homes, and the final destination of empty containers (they burn them, bury them, or throw them in with conventional garbage) (Table 7).

### 3.7. Damage to Health from Exposure to Pesticides

The average number of APP symptoms in the PPS was statistically higher than in the CPS (PPS: 7.0 ± 3.9, amplitude = 0–13, CPS: 4.1 ± 4.1, amplitude = 0–17, Mann–Whitney U = 208.0, *p* = 0.005). Burning and irritation of the skin predominated, followed by burning eyes, itchy skin, runny nose, headache, and mainly watery eyes (Figure 4). In general, 50.9% of users reported 1 to 5 APP symptoms, 27.3% 6 to 10 symptoms, and 14.5% >11 symptoms. A total of 7.3% of users stated that they did not present symptoms. No significant correlation was observed between the number of APP symptoms and the number of active ingredients used (Spearman’s Rho: 0.190, *p* = 0.164), nor with the time to fumigate one hectare (Spearman’s Rho: 0.129, *p* = 0.346). Still, a highly significant correlation was detected with the planted area (Spearman’s Rho: 0.312, *p* = 0.020).

## 4. Discussion

The crops documented are the main activity that generates economic income in the localities studied, allowing producers to support their families despite the region’s strong intermediary in marketing products. Both crops require a lot of labor force investment concerning the annual agricultural calendar, similar to what has been reported for Brazil and Indonesia [29,30], because they are susceptible and fragile crops. Therefore, they require much attention to prevent and control nutritional imbalances and attack pests and diseases. Both systems have been maintained under a conventional system through an essential diversity of industrial fertilizers and pesticides, as in many Latin American countries [31]. The differences in age, length of activity, and agricultural surfaces observed are because papaya is a product that generates greater economic profits, a situation that attracts the younger population to create and accumulate capital in the short and medium term. This is consistent with the findings of Girdziute et al. [32], even though the investment required is much more significant compared to chili.

Producers of both systems use various active ingredients with different commercial presentations. The justification for using these products is the high dependence on these inputs that producers have to prevent or control the various pests that attack their crops. Without them, the current productivity would not be achieved, which would put the economic investment they make at risk, as Popp et al. noted [33]. To ensure said productivity in both crops, weed control is carried out through the application of herbicides, which guarantees the adequate growth and development of the plants, as well as reducing the workload and time of agricultural cleaning tasks [34]. During fruiting, fumigation sessions are increased to control insects and diseases, guaranteeing that the fruits have a good appearance and can be easily marketed. The pests identified in both systems coincide with what has been reported in similar systems in other countries [35,36]. The strong presence of these pests is evidence of the ecological imbalance that has been occurring in recent years in these systems, as is happening in African agricultural systems [37].

Regarding the main active ingredients and most frequent chemical groups used, a different basic range of insecticides stands out for each crop. At the same time, the fungicide mancozeb and the herbicides paraquat and glyphosate are common in both systems. The main pesticides used in chili cultivation are similar to those reported in other countries [38], and those used in papaya coincide with what was reported for communities in Veracruz, Mexico [39]. It is worrying to have identified 33 active ingredients out of 49 classified as Highly Hazardous Pesticides [27], mainly due to their high acute toxicity, possible chronic effects on human health, and environmental toxicity. In addition, the majority are prohibited in many countries [28]. Due to their chronic effects, glyphosate, glufosinate ammonium, mancozeb, and captan stand out. Due to their high acute toxicity and/or environmental toxicity, paraquat, imidacloprid, lambda-cyhalothrin, beta-cyfluthrin, and abamectin stand out, all widely used in the systems studied.

In addition to the above, paraquat, methamidophos, and difocol are considered to have restricted use by COFEPRIS in Mexico [25]. Among the most relevant, given its controversy, is paraquat, since it is prohibited in many countries due to its extreme acute toxicity due to poisoning and exposure [40], and glyphosate due to its chronic toxicity (carcinogenic and genotoxic capacity) [41]. Unfortunately, many of them are widely distributed in many countries, mainly Asia and Latin America, including Mexico, despite their danger to human health and the environment since their residues and metabolites generate a negative impact on beneficial flora and fauna present in the ecosystems where they are applied [42]. The presence of these pesticides in the locations studied indicates their high commercialization in the region, most likely because they are cheap and readily available in their various commercial and generic presentations.

It is worth mentioning that producers are exposed to AIs individually, as well as to arbitrary mixtures of various AIs and toxicological categories. These combinations could generate synergies and increase the toxicity levels of one or more of the AIs used [43], in addition to indicating the high complexity of exposure and health risks to which farmers are subjected so the symptoms of damage to health can be multiple and manifest in the medium and/or long term. This generates warning signs, since short-term health damage was identified through the diversity of APP symptoms manifested by most users, accepting that several of the AIs they use may be responsible for said symptoms. In this regard, a considerable group of organophosphates and carbamates exists so that the APP symptoms could be related to the toxic properties associated with this group of pesticides [44,45]. The diversity of APP symptoms identified is similar to those reported in small rural producers in Costa Rica [46], Iran [47], and Thailand [48], as well as in the Chiapas Highlands [49] and recently in Oaxaca, Mexico [50]. Despite not having found a significant association between the number of APP symptoms and the number of AIs, the possibility that the perceived damages are derived from exposure to the identified AIs is not ruled out since the same producer uses several AIs at the same time, in addition to the fact that it could be strengthening a scenario conducive to the appearance of various chronic poisonings since several of the most frequently used pesticides have carcinogenic, neurotoxic, and teratogenic properties [51,52]. It has been reported that deltamethrin users have developed chronic lymphocytic leukemia, while glyphosate users have developed diffuse large B-cell lymphoma [53]. Some pyrethroids, such as bifenthrin, have accelerated the onset of neurodegenerative diseases in rats [54], while neonicotinoid insecticides (such as acetamiprid, imidacloprid, and thiamethoxam) are hepatotoxic [55].

Regarding the knowledge and behavior in the use and management of pesticides, it was observed that although the producers know how to read and write, they present inappropriate knowledge and behavior, which shows the low perception of risk in the management of pesticides concerning the impacts on their health and the environment, as happens in the agricultural region of Yucatán, Mexico [56]. In addition to the above, there is a notable lack of training and/or technical advice on the appropriate use and safety measures in the correct handling of pesticides by the companies that market these inputs, in addition to the evident disinterest of government entities state and municipal governments in addressing the problem, as happens in other countries [57]. The frequent days of fumigation in hot environments involved in a highly demanding agricultural calendar, as well as the lack of knowledge, little understanding, and/or erroneous beliefs about safety, hygiene, and personal protection practices, are factors that are related to the poor ability of farmers to protect themselves and increase the levels of exposure to pesticides. The above generates a high-risk environment that affects public health and the environmental environment of the locations studied despite the economic benefits generated by such agricultural activities. 

In Oaxaca, and particularly in the regions where the studied localities are located, there is little regulation of the use of pesticides. Although there is a national legal framework for regulating toxic substances in Mexico [58], it is not applied in its entirety and the authorities need to provide adequate attention to this problem. Consequently, the action of Mexican government institutions in regulating these products has been incipient. Recently, the federal government has attempted to regulate the use of glyphosate with initiatives aimed at agroecological alternatives; some results have been promising [59]. Unfortunately, as documented in this study, a large part of the Mexican agricultural sector is immersed in a conventional system highly dependent on fertilizers and pesticides. 

There are many challenges that the Mexican federal government has to face collectively through its different policies, programs, and sectors in agricultural, economic, health, and environmental matters, as well as those related to the commercial regulation of pesticides. It is highly advisable to promote comprehensive strategies regarding the correct use and management of pesticides and protective equipment; education, prevention, health care, and risk perception in agricultural environments, as well as the adoption of agroecological and/or organic strategies for food production, to promote healthier agricultural work environments and reduce the risk to public health.

On the other hand, despite the small sample size and the fact that no clinical parameters were determined to determine the damage to health, the findings related to APP symptoms derived from exposure to a variety of AIs are relevant data that could be used to build and strengthen future studies that contemplate techniques that allow the determination of biological markers to rule out possible damage or alterations in cell lines or functioning of various organs and/or systems or allow establishing a cause–effect relationship derived from exposure to certain specific pesticides in exposed individuals. It is necessary to implement a health surveillance program to identify the risks related to acute and chronic poisoning through relevant clinical tests that are accessible to the population in rural environments. 

Finally, it is pertinent to strengthen both production systems with training programs in the safe handling of pesticides, prevention and adequate control of pests, correct use of fertilizers, and the comprehensive use of natural resources, among other things, that promote agricultural productivity and improve economic returns without exposing the health of the rural population, as is currently happening.

## 5. Conclusions

The production systems studied require a lot of economic investment and a large labor force, mainly in papaya cultivation. Both systems depend highly on industrial inputs, primarily pesticides, for preventing and controlling insects and diseases. Most of the IAs identified are insecticides and fungicides of toxicological category IV, which are considered highly dangerous pesticides. They are applied individually or in arbitrary mixtures without the appropriate knowledge and behavior in their management, generating a diversity of APP symptoms among users, mainly burning and irritation of the skin, burning eyes, itchy skin, runny nose, headache, and watery eyes. These agricultural activities are relevant for generating economic income in the locations studied, but they put the health of the producers and their families at risk. Attention to this problem is necessary since both systems are potential scenarios for the manifestation of diseases due to exposure to highly dangerous pesticides in the medium and long term.

## Figures and Tables

**Figure 1 ijerph-21-01061-f001:**
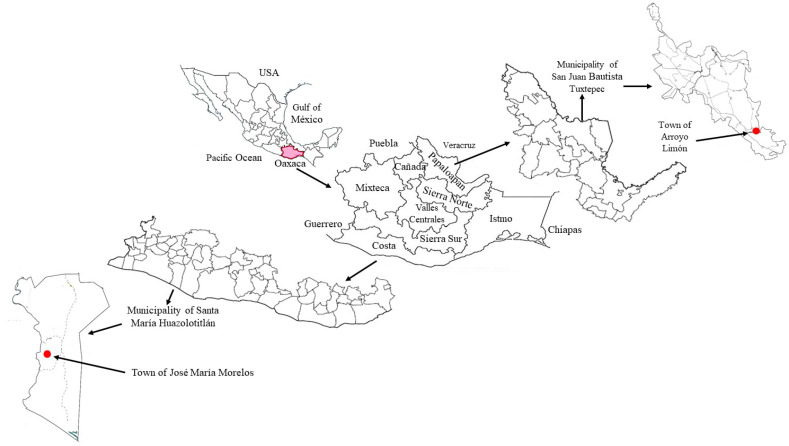
Location of the studied localities.

**Figure 2 ijerph-21-01061-f002:**
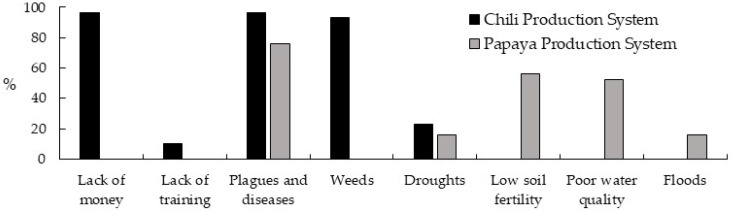
Problems in the production systems studied. The abscissa axis indicates the problem’s name, and the ordinate axis is the corresponding percentage.

**Figure 3 ijerph-21-01061-f003:**
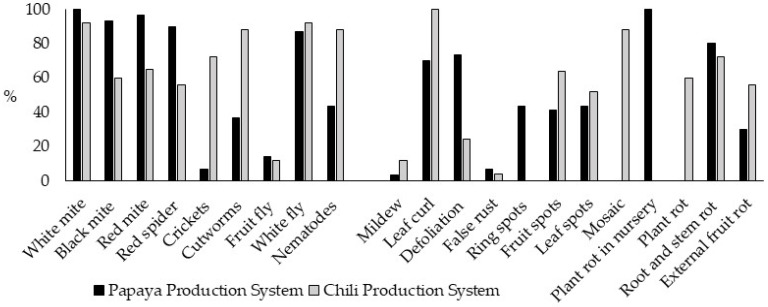
Presence of pests and diseases in the production systems studied. The abscissa axis indicates the insect or disease’s name and the ordinate axis shows the corresponding percentage.

**Figure 4 ijerph-21-01061-f004:**
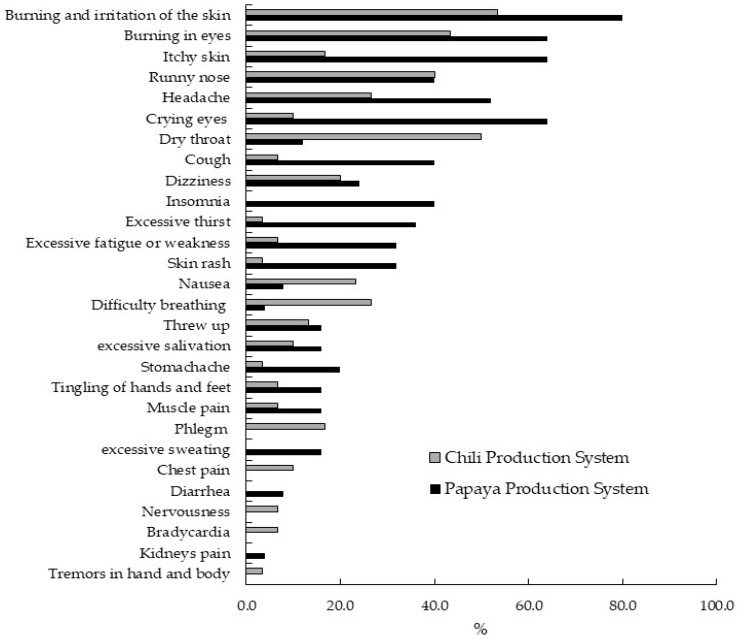
Symptoms of acute pesticide poisoning perceived by users. The ordinate axis indicates the symptom’s name, and the abscissa axis shows the corresponding percentage.

**Table 1 ijerph-21-01061-t001:** Main sociodemographic characteristics of the study population.

	PPS (n = 25)	CPS (n = 30)	Total (n = 55)	
Female/Male (%) *	0.0/100	3.3/96.7	1.8/98.2	X^2^ = 0.849, *p* = 0.357 ^NS^
Can read/Can’t read (%)*	100/0.0	93.3/6.7	96.4/3.6	X^2^ = 1.730, *p* = 0.188 ^NS^
Age (years) **	32.1 ± 9.9 (18–54)	41.9 ± 10.7 (26–62)	37.5 ± 11.4 (18–62)	t = −3.5, gl = 53, *p* = 0.001 ***
Seniority in agricultural work (years) **	12.0 ± 6.0 (5–30)	20.5 ± 9.6 (5–40)	16.7 ± 9.1 (5–40)	MWU = 571.5, *p* = 0.001 ***
Sown agricultural area (hectare) **	1.2 ± 0.7 (0.5–3.0)	0.8 ± 0.5 (0.2–2.0)	0.9 ± 0.6 (0.2–3.0)	MWU = 230.5, *p* = 0.011 ***

* X^2^ test. ** mean ± standard deviation (amplitude), Student’s *t*-test (t), or Mann–Whitney U (MWU), according to normality (Shapiro test) and homoscedasticity of variances (Levene test) of the data. PPS: papaya production system and CPS: chili production system. *** *p* < 0.05; NS = not significant.

**Table 2 ijerph-21-01061-t002:** Activities during the agricultural cycle in the production systems studied.

Activity	Chili Production System	Papaya Production System
Land selection	It is carried out 20 days before sowing, and lands with black clay-sandy soils are identified.	It is carried out a month before the transplant; flat areas with moderate sandy-clay textures are preferred, with little sludge formation and good drainage.
Seed selection	Pest-resistant plants (strong and large) are selected, and their fruits are cut and left in storage until they are fully ripe. Subsequently, the seeds are extracted and washed, immediately placed in a mosquito net bag for two hours, then spread on a plastic or cardboard surface to finish drying at room temperature. The clean seed is left to rest for approximately three months.	Vigorous plants are selected, and seeds are obtained from their fruits. The fruits are placed in water for one day and subsequently exposed to the sun for three days on a cloth surface, which must be permanently moist to promote roots. The seeds with the most roots are placed in the seedbed.
Preparation of the seedbed	It is carried out after the first or second rain of the year. On flat land close to the plot where the transplant will be carried out, a furrow 1.5 m wide and 2 m long is formed, and the surface layer is loosened and disinfected with lime, allowing it to rest for 1 to 5 days. Subsequently, the seeds are sown in the furrow, covering it with a plant cover (palm or dry grass) for eight days. Subsequently, the cover is raised to a height of 1 m to protect the seedbed, which is removed after 20 days when there are leaves on the seedlings. The corresponding transplant is carried out between 30 and 40 days after starting the seedling.	The seeds with the best roots are placed in the compartments of the trays (capacity of 200 compartments), which contain a substrate previously mixed with water for propagating seedlings in the greenhouse. They are allowed to grow until the seedlings are strong enough for transplanting.
Land preparation	Depending on the vegetation type, it may be cleaned with machetes and chainsaws. Some producers fallow the land, and others only harrow it. Then, lime is applied for disinfection. This activity is carried out before the transplant.	Before transplanting, the land must be tilled with the help of a tractor, disc plow, harrows, edger, and trencher. The disc plow loosens the soil and eliminates weeds by incorporating them into the soil. This activity forms mounds, which are removed by harrowing to even out the terrain. Immediately through the edger, the furrows are formed, and the planting is done by transplanting.
Transplant	In the previously selected land, 2 m wide alleys are made. In the middle of each alley and at a separation of 0.5 m, holes are made using a punch or spoke, where the seedlings previously submerged in a root-promoting solution are placed. This activity occurs in the morning (7:00 to 9:00 a.m.) or afternoon (4:00 to 6:00 p.m.).	One day before transplanting, a soil sealer is applied (to inhibit the growth of weeds in the first 20 days) and watered (to ensure soil moisture) so that the transplant can be carried out the next day. Two people carry out the activity. The first digs holes where he places one or two seedlings, and the second applies a rooting agent and an insecticide.
Reseeding	After a week after transplanting, the withered plants are replaced.	It consists of replacing those who became ill or did not resist the transplant with new plants. The eliminated plants are placed in a common grave, and lime is added. Replanting is carried out until the plant is five months old.
Sexing	This activity is not carried out.	Plants are selected according to their reproductive organ. The hermaphrodite plant is preferred. This process is carried out between the third and fourth month of the plant’s life, considering that, if replanting is necessary, the same sexing will also be carried out.
Irrigation system	The irrigation system is not used since the crop is seasonal.	PVC pipe and drip irrigation tape are installed and connected to a gasoline or electric pumping system. Irrigation begins once the transplant has been carried out. Subsequent irrigation will depend on humidity conditions and is carried out whenever necessary.
Preparation of trenches	This activity is not carried out.	Three months after the transplant, an agricultural trencher is used to carry out the work. A drainage channel is made between furrows, at a distance of 1.80 m, to form a drain, which serves to drain rainwater and transport the fruit using wheelbarrows to the washing, wallpapering, and stowage area.
Fertilization	Granular fertilizers are applied manually, and foliar fertilizers are applied by spraying. Five fertilizations are carried out during the cycle. In the first three days after transplanting (dat), various products are used: rooting agents by spraying, granules, and foliar. The second is achieved with foliar (5 dat); the third (20 dat) and fourth (30 or 40 dat) with granules; and the fifth with foliar (40 or 45 dat). Once the plants reach maturity and begin fruit production, foliar applications are applied twice per cut: the first when the producer considers it appropriate, and the second 10 days after the first application.	Two types of fertilization are carried out: soluble and granulated, and both are repeated every 20 days starting the month after transplanting. The first is the mixture of soluble fertilizers prepared in a 200 L drum, connected by hoses, and dispersed through drip irrigation to the crop. The second consists of the manual application of different solid fertilizers, which are mixed in a bucket, and the producer, using the palm of his hand, takes a portion and places it next to the base of each plant. Each producer chooses the fertilizer to apply based on their experience.
Pesticide application	Insecticides are applied from the seedling stage. At 15 dat, different herbicides are used. During the crop’s growth and development stage, insecticides and fungicides are applied. From the fourth application of fertilizers, they are regularly mixed with various pesticides. Spray pumps and/or motor pumps are commonly used, depending on the producer’s economic capacity.	Herbicides are applied at least every month throughout the life of the plant. Insecticides are used thrice a month from the second month to the tenth or eleventh months of life. For fungicides, they are applied from the third month of life (beginning of flowering) until the end of the cycle. Insecticides and fungicides are generally used in combination with foliar fertilizers, hormones, and bactericides. Their application is intensified during the flowering season (between the third and sixth months). When the plants are 8 to 9 months old, fumigation is carried out with motor or manual spray pumps.
Court	The first cut is made at 45 or 50 dat, subsequently every 20 days, up to a maximum of 3 or 4 cuts depending on the strength achieved in the plants with fertilization and pest control. This activity is carried out manually, and day laborers are generally hired. The harvest ends in January.	It is carried out after seven months of the plant’s life. The cutting is completed by hand. The fruit is placed in wheelbarrows (capacity approx. 150 kg) and transferred to the washing area, where it is immersed in a mixture of fungicides and dispersants. After washing, the fruit is wrapped in paper and stowed in a truck. The cut has a period of 5 to 6 months.
Commercialization	The product is transported to the community of Betania, municipality of Tuxtepec, where it is sold to intermediaries from Puebla or CDMX. The price per kg ranges between MXN 5 and 10.	The product is sold mainly to intermediaries established in the town, who transport it to the main markets of Puebla and CDMX. The price per kg ranges between MXN 6 and 11.
Final destination of post-harvest waste	At the end of the cycle, the plants are already too contaminated. The crop is left wholly abandoned for a month so that the weeds can grow. Then, the land is cleaned, and all the waste is burned. This strategy is used to prepare the land for the following agricultural cycle.	At the end of the cycle, the plants are approximately 2 m high, very sick, and no longer have enough foliage to cover the fruit. Therefore, they are cut down with a machete, and the stubble and remaining fruits are left on the plot to feed the livestock. Finally, the land is allowed to rest for approximately one year. some producers plant sorghum and/or corn for self-consumption during that time interval.

**Table 3 ijerph-21-01061-t003:** Agricultural calendar of the production systems studied.

January	February	March	April	May	June	July	August	September	October	November	December
				Land selection						Seed selection	
					Land preparation						
					Preparation of the seedbed						
					Insecticide application						
							Transplant				
							Fertilization (1st, 2nd and 3rd)				
							Herbicide application				
								Fertilization (4th)			
								Application of insecticides and fungicides			
								First cut			
									Fertilization (5th)		
									Second cut		
										The third or fourth cut	
Commercialization									Commercialization		
Harvest term	The final destination of the plant										
								Land selection			
									Seed selection		
									Preparation of the seedbed		
									Land preparation		
									Irrigation system installation		
									Transplant		
Irrigation										Irrigation	
Fertilization										Fertilization	
Herbicide application										Herbicide application	
Reseeding										Reseeding	
	Ditched										
Application of insecticides and fungicides											
Sexing											
				Court							
				Commercialization							
										The final destination of the plant	

Cells in yellow are activities of the Chili Production System, and cells in blue are activities of the Papaya Production System. Some activities begin at the beginning of the corresponding month; others start in the second, third, or fourth week of the indicated month.

**Table 4 ijerph-21-01061-t004:** Classification of pesticides identified by CT.

TC	Insecticides Number AI (CP)		Herbicides Number AI (CP)		Fungicides Number AI (CP)		Bactericides Number AI (CP)		Total Number AI (CP)	
	PPS	CPS	PPS	CPS	PPS	CPS	PPS	CPS	PPS	CPS
I	1 (1)	2 (2)	--	--	--	--	--	--	1 (1)	2 (2)
II	4 (4)	5 (5)	--	--	--	--	--	--	4 (4)	5 (5)
III	4 (4)	4 (4)	2 (6)	2 (7)	--	--	--	--	6 (10)	6 (11)
IV	5 (6)	11 (16)	2 (4)	2 (7)	11 (13)	7 (8)	--	--	18 (23)	20 (31)
V	--	1 (1)	--	--	--	1 (1)	2 (1)	2 (1)	2 (1)	4 (3)
Total	14 (15)	23 (28)	4 (10)	4 (14)	11 (13)	8 (9)	2 (1)	2 (1)	31 (39)	37 (52)

TC = toxicological category (based on the most frequently used commercial product): I (extremely dangerous), II (highly dangerous), III (moderately dangerous), IV (slightly hazardous), and V (usually not dangerous) [25,26]. AI (CP) = active ingredient (commercial product). PPS = papaya production system, CPS = chili production system.

**Table 5 ijerph-21-01061-t005:** Pesticides identified by chemical classification and hazard.

Active Ingredient	Chemical Group	Use Class ^(a)^	TC ^(b)^	Number of Users Using the Commercial Product (PPS: n = 25/CPS: n = 30)	HHP List ^(c)^	Total AI Bans World- Wide ^(d)^	% of Users	
							PPS n = 25	CPS n = 30
Paraquat *	Bipyridyl	H	III	Gramoxone (18/23) Cuproquat (1/--) Mata todo (3/--) Lumbrequat (3/--) Ojiva (1/--) Paraquat (--/9) Gramoxil (--/8) Diabloquat (--/1) Dragocson (--/1) Cerillo (--/1)	1, 4	58	100	100
2, 4-D	Chlorophenoxy (phenoxy acetic acid)	H	III	Herbipol (2/1)	2	5	8.0	3.3
Glyphosate	Phosphonomethylglycine (phosphonate)	H	IV	Glyphosate (20/16) Machete (2/--) Rondo (5/--) Faena (--/3) Lafam (--/3) Velfosato (--/3)	2	4	92.0	83.3
Ammonium glufosinate	Organophosphate	H	IV	Finale (5/9) Tarang (--/10) Glufosinate (--/1)	2	29	20.0	66.6
Aluminum phosphide	Inorganic phosphide	I	I	Fosfusan (--/4)	1, 3	2	--	13.3
Methyl parathion	Organophosphate	I	I	Foley (6/4)	1, 4	--	24.0	13.3
Carbofuran	Carbamate	I	II	Furadan (10/3)	1, 3, 4	87	40.0	9.9
Endosulfan	Organochlorine	I	II	Thiodan (2/--)	1, 4	130	8.0	--
Fipronil	Phenylpyrazole	I	II	Regent 4 SC (--/3)	3	38	--	9.9
Methamidophos *	Organophosphate	I	II	Tamarón (1/1)	1, 3, 4	109	4.0	3.3
Methomyl	Carbamate	I	II	Lannate (19/7)	1, 3	47	76.0	23.3
Oxamyl	Carbamate	I	II	Vydate (--/2)	1, 3	5	--	6.6
Bifenthrin	Pyrethroid	I	III	Talstar técnico (4/--)	2, 3	30	16.0	--
Chlorpyrifos ethyl	Organophosphate	I	III	Lorsban 50 W (--/2)	2, 3	39	--	6.6
Dimethoate	Organophosphate	I	III	Rogor (2/--)	3	3	8.0	--
Lambda-cyhalothrin	Pyrethroid	I	III	Kirio (2/--)	1, 3	--	8.0	--
Lambda-cyhalothrin + Tiametoxam	Pyrethroid/Neonicotinoid	I	III/IV	Engeo (--/16)	1, 3/3	--/28	--	53.3
Permethrin	Pyrethroid	I	III	Ambush 340 (3/--) Permethrin (--/1)	2, 3	33	12.0	3.3
Thiodicarb + Imidacloprid	Carbamate/Neonicotinoid	I	III/IV	Semevin xtra (--/2)	2, 3/3	33/29	--	6.6
Abamectin	Avermectin	I	IV	Abamectin 1.8% (--/6) Agrimec (--/2) Agriver (--/1) Lucatina (--/1)	1, 3	--	--	33.3
Abamectin + Spirodiclofen	Avermectin/Derived from tetronic acid	I	IV/V	Envidor speed (--/2)	1, 3/2	--/29	--	6.6
Acetamiprid	Neonicotinoid	I	IV	Rescate (1/--)	--	--	4.0	--
Cypermethrin	Pyrethroid	I	IV	Crucial 20 CE (1/--) Arrivo (--/3)	3	29	4.0	9.9
Cyromazine	Triazine	I	IV	Trigard (--/3)	--	--	--	29.9
Dicofol *	Organochlorine	I	IV	AK-20 (--/3)	4	52	--	9.9
Flupyradifuran	Butenolide	I	IV	Sivanto (--/1)	3	--	--	3.3
Imidacloprid	Neonicotinoid	I	IV	Confidor (3/22) Bration (1/--) Picador (--/2)	3	29	16.0	73.3
Imidacloprid + Beta-cyfluthrin	Neonicotinoid/Pyrethroid	I	IV/IV	Muralla Max (5/13)	3/1, 3	29/30	20.0	43.3
Malathion	Organophosphate	I	IV	Malathion (8/--)	2, 3	32	32.0	--
Spiromesifen	Derived from tetronic acid	I	IV	Oberón (--/8)	--	--	--	26.6
Spirotetramac	Derived from tetronic acid	I	IV	Movento 240 (--/2)	--	--	--	6.6
Spirodiclofen	Derived from tetronic acid	I	V	Envidor (--/3)	2	29	--	9.9
Azoxystrobin	Strobirulins	F	IV	Amistar (10/--)	--	--	40.0	--
Azoxystrobin + Difeconazole	Strobirulins + Triazole	F	IV/IV	Amistar Gold (1/--)	--/--	--/1	4.0	--
Benomyl	Benzimidazole	F	IV	Promyl (--/4)	2, 3	39	--	13.3
Captan	Carboxamide	F	IV	Captan 50 WP (11/--) Captan (--/10)	2	6	44.0	33.3
Chlorothalonil	Chloronitriles	F	IV	Clorimex 500 F (2/--)	1, 2	34	8.0	--
Cymoxanil + Mancozeb	Cyanoacetamide oxime + Dithiocarbamate	F	IV/IV	Curzate M8 (--/2)	--/2	--/31	--	6.6
Carbendacim	Bencimidazole	F	IV	Prozycar (--/1)	2	34	--	3.3
Mancozeb	Dithiocarbamate	F	IV	Manzate (19/17)	2	31	76.0	56.6
Metalaxyl	Phenylamide	F	IV	Metalaxyl (8/4) Tokat 240 (3/--)	--	1	44.0	13.3
Metalaxyl + Chlorothalonil	Phenylamide/Chloronitriles	F	IV/IV	Malakal (2/--)	--/1, 2	1/34	8.0	--
Metalaxyl-M + Mancozeb	Phenylamide/Dithiocarbamate	F	IV/IV	Ridomil Gold MZ 68 (--/5)	--/2	1/31	--	16.6
Prochloraz	Imidazole	F	IV	Sportak 45 CE (12/--)	--	29	48.0	--
Pyraclostrobin + Boscalid	Strobirulin/Carboxamide	F	IV/IV	Cabrio C (1/--) Cabrio (1/--)	--/--	--/--	8.0	--
Tebuconazole	Triazole	F	IV	Tebuzan (--/4)	1, 2	1	--	13.3
Trifloxystrobin	Strobirulins	F	IV	Tega 500 SC (1/--)	--	--	4.0	--
Fosetyl Al	Organophosphate	F	V	Aliette (--/1)	--	--	--	3.3
Streptomycin + Oxytetracycline	Antibiotic/Antibiotic	B	V/V	Agri-mycin 100 (--/4)	--/--	30/29	--	13.3
Streptomycin + oxytetracycline + Copper oxychloride	Antibiotic/Antibiotic/Inorganic copper compounds	B/B/F	V/V/IV	Agrimycu 500 (1/--)	--/--/--	30/29/--	4.0	--
Oxytetracycline	Antibiotic	B	V	Terramycin 5% (--/1)	--	29	--	3.3

^(a)^ H/I/F/B: herbicide/insecticide/fungicide/bactericide. ^(b)^ TC: toxicological category (based on the most frequently used commercial product): I (extremely dangerous), II (highly dangerous), III (moderately dangerous), IV (slightly hazardous), and V (usually not dangerous) [25,26]. ^(c)^ Criteria for inclusion in the PAN International Highly Hazardous Pesticides list: [1] high acute toxicity; [2] chronic effects on human health; [3] environmental toxicity; and [4] restricted or prohibited by environmental conventions [27]. ^(d)^ PAN International consolidated list of banned pesticides [28]. * Use restricted by COFEPRIS [25].

**Table 6 ijerph-21-01061-t006:** Description of the identified pesticide mixtures.

TC II AI (Type of Pesticide) Commercial Product	TC III AI (Type of Pesticide) Commercial Product (No. of Users)	TC IV AI (Type of Pesticide) Commercial Product (No. of Users)	% of Users
**Papaya Production System (n = 25)**			
Carbofuran (I) + Furadan + Furadan +	Paraquat (H) Matatodo (2) Gramoxone (1)		12.0
Methamidophos (I) + Tamarón +	Paraquat (H) Gramoxone (1)		4.0
	Dimethoate (I) + Paraquat (H) Rogor + Gramoxone (1)		4.0
	Paraquat (H) + Lambda-cyhalothrin (I) Gramoxone + Kirio (2)		8.0
	Paraquat (H) + Ojiva + Gramoxone +	Malathion (I) Malathion (1) Malathion (3)	12.0
	Lambda-cyhalothrin (I) + Kirio + Kirio +	Glyphosate (H) Glyphosate (2) Rondo (2)	16.0
		Glyphosate (H) + Malathion (I) Rondo + Malathion (2) Glyphosate + Malathion (2)	16.0
Carbofuran (I) + Furadan +		Glyphosate (H) Glyphosate (3)	12.0
Carbofuran (I) + Furadan +		Ammonium glufosinate (H) Finale (2)	8.0
Methamidophos (I) + Tamarón +	Dimethoate (I) Rogor (1)		4.0
Methomyl (I) + Lannate +		Prochloraz (F) Sportak 45 CE (6)	24.0
Methomyl (I) + Lannate +		Pyraclostrobin/Boscalid (F) Cabrio (1)	4.0
		Imidacloprid + Beta-cyfluthrin (I) + Captan (F) Muralla Max 300 + Captan (1)	4.0
		Prochloraz (F) + Mancozeb (F) Sportak 45 CE + Manzate (3)	12.0
		Metalaxil (F) + Mancozeb (F) Metalaxil + Manzate (1)	4.0
		Chlorothalonil (F) + Mancozeb (F) Clorimex 500 F + Manzate (1)	4.0
		Captan (F) + Mancozeb (F) Captán + Manzate (3)	12.0
		Trifloxystrobin (F) + Prochloraz (F) Tega 500 SC + Sportak 45 CE (1)	4.0
		Captan (F) + Pyraclostrobin/Boscalid (F) Captán + Cabrio C (2)	8.0
		Mancozeb (F) + Pyraclostrobin/Boscalid (F) Manzate + Cabrio (1)	4.0
**Chili Production System (n = 30)**			
		Ammonium glufosinate (H) + Glyphosate (H) Finale + Velfosato (1)	3.3
	Paraquat (H) + Gramoxone + Paraquat + Dragocson +	Ammonium glufosinate (H) Fínale (1) Tarang (3) Tarang (1)	16.6
	Paraquat (H) + Gramoxone + Gramocil + Gramoxone + Paraquat +	Glyphosate (H) Glyphosate (1) Lafam (1) Faena (1) Glyphosate (1)	13.3
		Spiromesifen (I) + Imidacloprid (I) Oberon + Confidor (1)	3.3
		Abamectin (I) + Benomilo (F) Abamectin + Promyl (1)	3.3
		Captan (F) + Benomyl (F) Captan + Promyl (1)	3.3
Carbofuran (I) + Furadan +		Mancozeb (F) + Carbendacim (F) Manzate + Prozicar (1)	3.3
		Mancozeb (F) + Benomyl (F) Mancozeb + Promyl (1)	3.3
		Mancozeb (F) + Streptomycin + Oxytetracycline (B) Mancozeb + Agri-mycin 100 (1)	3.3
		Mancozeb (F) + Metalaxyl (F) + Benomyl (F) Manzate + Metalaxyl + Promyl (1)	3.3
	Chlorpyrifos ethyl (I) + Lorsban 50 W +	Mancozeb (F) Manzate (1)	3.3
	Lambda-cyhalothrin (I)/Tiametoxam (I) + Benomyl (F) Engeo + Promyl (1)		3.3
Fipronil (I) + Regent 4 SC +	Lambda-cyhalothrin (I)/Tiametoxam (I) Engeo (1)		3.3

TC = toxicological category. AI (type of pesticide) = active ingredient (H/I/F/B: herbicide/insecticide/fungicide/bactericide).

**Table 7 ijerph-21-01061-t007:** Positive responses to knowledge and appropriate behaviors in the use and management of pesticides.

	PPS % (n = 25)	CPS % (n = 30)
1. Have you received training in the use and management of pesticides?	40.0	32.0
2. Did you learn how to use pesticides from professionals?	10.0	0.0
3. Does the person who sells you pesticides tell you how to use them?	66.7	76.0
4. When you buy a pesticide for the first time, do you read the label?	83.3	36.0
5. Do you know what the colors on pesticide labels mean?	0.0	12.0
6. When preparing the pesticide for fumigation, do you follow the instructions on the label?	30.0	24.0
7. Do you use personal protective equipment to cover your hands when preparing pesticides for fumigation?	13.3	4.0
8. Are there no spills or splashes when preparing the pesticide for fumigation?	26.7	8.0
9. Do you wash your hands after preparing the pesticide for fumigation?	96.7	100.0
10. Is your spray pump in good condition?	6.7	100.0
11. When the pump nozzle becomes clogged, do you use any tool to unclog it?	6.7	12.0
12. After spraying, do you wash your hands before eating or doing any other activity?	90.0	96.0
13. Do you bathe after fumigating?	96.7	96.0
14. Do you use special clothing to fumigate?	0.0	0.0
15. Do you use face masks when fumigating?	0.0	0.0
16. When it is very windy, do you avoid applying pesticides?	80.0	68.0
17. Do you wash the equipment when you finish fumigating?	26.7	52.0
18. Do you store pesticides in a particular place outside your home?	0.0	60.0
19. Do you place empty pesticide containers in special containers?	0.0	76.0

PPS = papaya production system, CPS = chili production system.

## Data Availability

The data presented in this study are available upon request from the corresponding author.

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
