# Peer review of "Pesticide Exposure in the Cultivation of *Carica papaya* L. and *Capsicum annuum* L. in Rural Areas of Oaxaca, Mexico"

_ijerph, 2024, doi:10.3390/ijerph21081061_

Round 1
Reviewer 1 Report
Comments and Suggestions for Authors
Manuscript ijerph-3131091 “Pesticide exposure in the cultivation of Carica papaya L. and Capsicum annuum L. in rural areas of Oaxaca, Mexico” focused on the Pesticide exposure in the cultivation of Carica papaya L. and Capsicum annuum L.. This research was important to human health. But this manuscript cannot be publication on International Journal of Environmental Research and Public Health with current form. The writing and research should be improved.
1. The abstract and keywords should be rewrite.
2. Has the data on pesticide residue exposure received by relevant personnel been analyzed and statistically analyzed when studying pesticide exposure?
3. The authors were suggested to supplement some relevant research results about pesticide residues.
Author Response
Manuscript ijerph-3131091 “Pesticide exposure in the cultivation of Carica papaya L. and Capsicum annuum L. in rural areas of Oaxaca, Mexico” focused on the Pesticide exposure in the cultivation of Carica papaya L. and Capsicum annuum L.. This research was important to human health. But this manuscript cannot be publication on International Journal of Environmental Research and Public Health with current form. The writing and research should be improved.
- The abstract and keywords should be rewrite.
- Has the data on pesticide residue exposure received by relevant personnel been analyzed and statistically analyzed when studying pesticide exposure?
- The authors were suggested to supplement some relevant research results about pesticide residues.
The summary was corrected. In the communities studied and, in general, for the state of Oaxaca, no studies have been carried out on pesticide residues in humans.
Reviewer 2 Report
Comments and Suggestions for Authors
Manuscript Review
Brief Summary
The manuscript "Pesticide Exposure in the Cultivation of Carica papaya L. and Capsicum annuum L. in Rural Areas of Oaxaca, Mexico" examines the use of pesticides, the knowledge and behaviors surrounding their application, and the acute poisoning symptoms (APS) experienced by farmers in two rural communities in Oaxaca. The study identifies 31 active ingredients (AIs) in papaya and 37 in chili cultivation, primarily insecticides and fungicides. The findings highlight the significant health risks to producers and their families due to inappropriate pesticide handling and exposure.
General Concept Comments
The study addresses a critical public health issue in agricultural communities, highlighting the widespread use of hazardous pesticides and the consequent health risks. Focusing on rural areas in Oaxaca provides valuable insights into the local farming practices and challenges small-scale farmers face.
The methodology is robust. Surveys are utilized to gather detailed sociodemographic information, pesticide usage, and health symptoms. The statistical analysis thoroughly compares differences between papaya and chili producers and examines correlations between pesticide exposure and health outcomes.
The manuscript is generally well-written and organized, with a clear structure and logical flow. However, some sections could benefit from additional clarity and detail, particularly in discussing results and their implications.
Specific Comments
Abstract: The abstract could be more straightforward about the main findings and their significance. It should mention particular health risks and the percentage of producers experiencing APS.
2. Introduction: Provide more background on the specific regions in Oaxaca and their significance in Mexican agriculture. Explain why these communities were chosen for the study. Also, more references should be included to existing literature on pesticide exposure and health impacts in similar agricultural settings.
3. Materials and Methods: The description of the study areas (José María Morelos and Arroyo Limón) is thorough, but consider improving the maps to enhance understanding (Figure 1). How were the questions developed, and were there any pre-tests conducted?
4. Results: Some tables and figures could be more effectively presented. Ensure all figures and tables are clearly labeled and referenced in the text. Also, please provide more details on the statistical tests used, including assumptions and justifications for their use. Explain why specific tests were chosen over others.
5. Discussion: Discuss the long-term health implications of pesticide exposure more thoroughly. What chronic conditions could arise from continued exposure?
6. Conclusions: The conclusions effectively summarize the key findings but could be more impactful by reiterating the most critical health risks and the urgency of addressing them.
Author Response
- Abstract: The abstract could be more straightforward about the main findings and their significance. It should mention particular health risks and the percentage of producers experiencing APS.
In response to the comment, the main symptoms of acute pesticide poisoning were included, and emphasis was placed on the health risks due to pesticide exposure. The percentage of producers who manifest APP symptoms is already mentioned and is underlined in yellow.
Introduction: Provide more background on the specific regions in Oaxaca and their significance in Mexican agriculture. Explain why these communities were chosen for the study. Also, more references should be included to existing literature on pesticide exposure and health impacts in similar agricultural settings.
More information was included about the importance of the crops studied in Mexican agriculture, emphasis was placed on the choice of study communities, and more information and references were included on pesticide exposure and its impact on health in similar agricultural environments in southeastern Mexico.
Materials and Methods: The description of the study areas (José María Morelos and Arroyo Limón) is thorough, but consider improving the maps to enhance understanding (Figure 1). How were the questions developed, and were there any pre-tests conducted?
The location map of the communities studied was improved, and it was clarified that the survey was piloted and validated by the study's researchers.
- Results: Some tables and figures could be more effectively presented. Ensure all figures and tables are clearly labeled and referenced in the text. Also, please provide more details on the statistical tests used, including assumptions and justifications for their use. Explain why specific tests were chosen over others.
It was reviewed that the figures and tables were adequately labeled and referenced in the text, and some annotations were made to make them more effective. The Materials and Methods section detailed the statistical tests used more clearly.
- Discussion: Discuss the long-term health implications of pesticide exposure more thoroughly. What chronic conditions could arise from continued exposure?
Discussion regarding long-term implications for health and diseases that may arise was included.
- Conclusions: The conclusions effectively summarize the key findings but could be more impactful by reiterating the most critical health risks and the urgency of addressing them.
Annotations were included to emphasize the health risks of chronic exposure to highly hazardous pesticides.